# Synergies with and Resistance to Membrane-Active Peptides

**DOI:** 10.3390/antibiotics9090620

**Published:** 2020-09-19

**Authors:** Adam Kmeck, Robert J. Tancer, Cristina R. Ventura, Gregory R. Wiedman

**Affiliations:** Department of Chemistry and Biochemistry, Seton Hall University, South Orange, NJ 07079, USA; adam.kmeck@student.shu.edu (A.K.); robert.tancer@student.shu.edu (R.J.T.); cristina.ventura@shu.edu (C.R.V.)

**Keywords:** membrane-active peptides, antimicrobial-resistance, drug synergy

## Abstract

Membrane-active peptides (MAPs) have long been thought of as the key to defeating antimicrobial-resistant microorganisms. Such peptides, however, may not be sufficient alone. In this review, we seek to highlight some of the common pathways for resistance, as well as some avenues for potential synergy. This discussion takes place considering resistance, and/or synergy in the extracellular space, at the membrane, and during interaction, and/or removal. Overall, this review shows that researchers require improved definitions of resistance and a more thorough understanding of MAP-resistance mechanisms. The solution to combating resistance may ultimately come from an understanding of how to harness the power of synergistic drug combinations.

## 1. Introduction

Membrane-active peptides (MAPs) are peptides ranging from about 4–40 amino acids in length that can interact with the cell membrane through permeabilization or other antimicrobial mechanisms [1,2]. They are often comprised of amino acid residues that are positively charged at pH 7. They can be grouped into four main structural categories: Linear α-helices, extended structures (usually abundant in Glycine, Arginine, Tryptophan, or Proline residues), β-sheets (often stabilized by disulfide linkages), and loops that contain both α and β moieties [3]. Table 1 shows examples of these peptides that are discussed in this review. In the past 10 years publication of materials concerning MAPs has exponentially increased [2]. This increase is partly due to MAPs potential ability to combat antimicrobial-resistance [4]. MAPs are found in numerous organisms ranging from humans (α-defensin) to insects (cecropin A) [2]. Extensive studies in this field have shown that there are many varied mechanisms of action. Proposed mechanisms include: The formation of toroidal and barrel stave pores, as well as non-pore forming mechanisms, such as carpet, detergent, inverted micelle, and membrane thinning models [1,2,5]. All these mechanisms contribute to the destabilization of lipid membranes in one manner or another.

Disruption of the cell membrane of a microorganism can have many deleterious effects. Biophysical studies have long supported the hypothesis that MAPs cause depolarization of the cell membrane, which removes the electrochemical gradient needed to drive molecular transport across the membrane [6,7]. An additional role may be played by the ability of MAPs to corral anionic molecules, specifically charged lipids, leading again to disruption of transport and disruption of signal transduction [8,9,10]. Total disruption of the membrane and escape of large, macromolecules has been seen and an important mechanism with synthetic MAPs [11]. Researchers in this area have repeated the claim that, no matter what the exact mechanism, the ability of MAPs to target the cell membrane allows them to either partially or totally circumvent the development of resistance [12,13,14]. In this review, we explore the literature surrounding resistance to MAPs, and we offer suggestions for ways forward to address those possible avenues of resistance.

In thinking about the way MAPs interact with the cell membrane, we would define possible resistance and synergy in four stages (Figure 1). The first stage would be when the peptides are still outside of the cell. At this point, they are subject to peptidases, which may degrade them [15]. The use of MAPs could select for organisms that have mutated extracellular peptidases, enhanced extracellular peptidase synthesis pathways, improved transport of peptidases to the extracellular space, or a combination. The second stage is at the cell wall or outer membrane of the cell. This structure often contains charged molecules that can bind to MAPs and prevent them from reaching the cell membrane. The third stage is when the peptide interacts with the lipids in the cell membrane. The assumption in designing or utilizing MAPs has been that the composition of the inner and outer leaflets of the cell membrane remains static overtime. The cell membrane, however, contains enzymes that constantly reshape lipid composition: flippases and flopases [16,17,18,19]. Organisms that can more rapidly alter the composition of their cell membrane leaflets via mutated flippases can better respond to a threat from MAPs, and hence, may develop resistance. Finally, when other small drugs do reach the membrane or intracellular space, they may be expelled by endogenous transporters [20,21,22]. This is already well accepted in the microbiology community as an emerging avenue of antimicrobial-resistance wherein MAPs may prove synergistic. In this review, we argue that to combat resistance by flippases, peptidases, charged structural components, and transporters, a method must be employed that looks for potential drug synergies that could specifically address those three avenues.

## 2. Defining Antimicrobial Synergy and Resistance

### 2.1. Minimum Inhibitory Concentration Assays

When discussing activity, it is important to try to define and quantify the terms synergy and resistance with respect to antimicrobial peptides. The common metric for comparing antimicrobial efficacy is the drug minimum inhibitory concentrations (MIC) and the Fractional Inhibitory Concentration (FIC) for drug combinations [23]. Above this concentration, no growth of a specific microbe will occur in standard 96-well microdilution or agar plate assays [24]. For MAPs these values can vary between 0.1 µg/mL to greater than 1 mg/mL [25,26]. It should be noted that for a direct comparison to small-molecule antimicrobials, a discussion of molar concentration may be more appropriate. The MAPs discussed in comparative studies generally have a mass of at least 1 kDa or greater [25]. As an example, the common antibiotic, Benzylpenicillin, with a molecular weight of 334 Da, has an MIC value of up to 0.06 µg/mL against certain bacterial strains [27]. Given this fact, it could be estimated that the efficacy of MAPs is at least two-fold and possibly more than ten-fold less than small-molecule antibiotics on a mole-for-mole basis.

Even though they may be less efficacious, the claim has always been that MAPs are less susceptible to resistance. Compared to β-lactams, there may be fewer mutant strains of microorganisms that can tolerate concentrations of MAPs above their reported minimum inhibitory concentrations. Groups, such as the Clinical Laboratory Standards Institute (CLSI) and the European Committee on Antimicrobial Susceptibility Testing yearly define “breakpoints” in microorganisms as the MIC values above which an organism may be called “resistant” to a given drug [28,29]. There are a myriad of challenges to consider when determining breakpoints for MAPs and seemingly no consensus on how to compare them [30]. The breakpoints are influenced by the exact MAP mechanism of action, as well as how susceptible they are to degradation. It is very clear, however, that breakpoints of MAPs are not nearly as widely discussed as small-molecule antimicrobials. 

### 2.2. Antimicrobial Activity Assays

When considering MAPs as antimicrobial agents, it is important to emphasize the fact that minimum inhibitory assays come with their own difficulties in terms of clinical translation. The efficacy of a peptide against a given microorganism may depend on the starting concentration of the microorganism in a sample leading to an inoculum effect [31]. This was seen in a study of the peptide pexiganan against *Escherichia coli* [32,33]. Among twelve replicates of the same 5 × 10^5^ cell/mL inoculum, an initial MIC value of 5 µM (~12.4 µg/mL) was determined with significant variability in well-to-well results. Further studies of time-kill curves and at sub-MIC values revealed that the peptide may only be able to kill a fraction of the bacteria at that concentration. The remaining bacteria are, therefore, able to grow as readily as those unexposed to pexiganan. The authors claim that peptide depletion over time, due to aggregation effects, accumulation of the drug in dead cell membranes, or adhesion to cellular components (such as DNA or charged proteins) are possible causes. This all suggests that defining MIC values for peptides is a challenging problem and that inoculum effects are important when considering the development of resistance.

Other methods of qualifying MAP efficacy exist beyond MIC assays. A recent comprehensive review of novel methods to determine antimicrobial susceptibility notes using techniques, such as microscopy, microfluidics, cytometry, genomic database searches, and others, to study drug efficacy [34]. A key advantage of many of these techniques is their ability to circumvent problems discussed above with MIC assays through better tracking of growth changes and morphology changes in the microorganism. Of particular note for MAPs are techniques involving model membrane systems in microfluidic high throughput platforms [35]. MAPs can be screened for their activity against a variety of well-defined model microbial membranes as giant unilamellar vesicles to provide some understanding of their activity. Taken together, MIC assays, along with newer susceptibility testing mechanisms, are crucially important for understanding the development of antimicrobial-resistance. The question then becomes, what can we do to understand and anticipate the ways in which microorganisms may develop resistance to antimicrobial peptides and what mechanisms of synergy exist? 

## 3. Synergy and Resistance to MAPs

### 3.1. Outside of the Cell

The first place to discuss resistance and synergy with regards to MAPs is outside of the cell. Gram-positive and gram-negative bacteria possess the ability to secrete extracellular proteases that can degrade MAPs and other peptide-based molecules [36,37]. This becomes paramount in some species like *Staphylococcus aureus* where these secreted proteases impact biofilm formation and disease progression [38]. Extracellular proteases are not only produced in bacterial species, but also in certain fungal species, such as *Aspergillus nidulans*. Branched Amphiphilic Peptide Capsules (BAPCs), which are MAP-like, peptide-based nanoparticles, are degraded through the interaction of extracellular proteases from this microorganism [39]. Clearly, degradation of MAPs in the extracellular space, before the molecules reach the cell surface, lowers their efficacy. This suggests that protease inhibition would promote synergy and lead to decreased MIC values.

Inhibition of extracellular proteases has become a viable route to achieve synergy with small-molecule drugs and could also show synergy with MAPs. As a notable example, there is a key serine protease, SspA/V8, that is secreted by *S. aureus* [40]. This protein is crucial to microbial growth and adhesion that has been shown to degrade the MAP LL-37 [41]. Inhibitors of the SspA/V8 endoprotease were discovered through genetic analysis of *Ruminococcus bromii*, which may ultimately show synergy with MAPs against *S. aureus* [42]. Streptococcal pyrogenic exotoxin B (SpeB), a cysteine protease, exhibits pathogenic activity along with degradation capability towards vital biological host proteins along with MAPs, such as LL-37 [43]. As one of the important virulence factors secreted by *Streptococcus pyrogenes*, SpeB is a promising target to inhibit, and inhibition has been achieved through the utilization of allicin and a 2S-alkyne. Analogous to the previously discussed extracellular proteases, the Omptin family, an outer membrane protease family, can also degrade MAPs [44]. Aprotinin has been shown to inhibit proteases from this family, such as CroP from *Citrobacter rodentium*, Pla from *Yersinia pestis*, and OmpT from *E. coli*, in order of effectiveness [45]. Fungal species, such as *Aspergillus fumigatus* and some from the genus *Candida*, are also infectious agents that secrete extracellular proteases [46,47,48]. Inhibition of secreted aspartic protease 2 (SAP2) has been seen with small molecules that not only inhibit biofilm and hypha formation, but also indicated strong efficacy in vivo [49]. The inhibition of these extracellular microbial proteases is an important source of synergy. 

In an interesting paradox, MAPs often plan an inhibitory role with respect to the host immune response. Certain MAPs are noted for their ability to impact common immune pathways involved inflammation [50,51]. Notably, MAPs have been shown to increase levels of interferon gamma (INF-γ), interleukin 10 (IL-10), and to reduce levels of tumor necrosis factor alpha (TFN-α) [52]. All of this suggests that the removal of MAPs from the extracellular space would improve the host immune response to an infection. Developing tolerance to high concentrations of MAPs or the ability to decrease the effective concentration below the MIC, thus, provides an even greater advantage to microorganisms. The effect of MAPs in this instance may be similar to nonsteroidal anti-inflammatory drugs; which is to exacerbate the infection if it is unable to completely kill the microorganism [53]. This makes maintaining an extracellular MAP concentration well above the MIC value important. 

An estimate of the extent of a protease inhibitor-MAP combination synergy can be derived by considering the plasma half-life of peptide drugs. Peptides are degraded quickly in plasma by mammalian proteases, their stability being, in some cases, on the order of minutes [54]. In practical cases, treatments for severe infections and sepsis may require antibiotic application over a period of hours [55,56]. Thus, if an adequate level of an intervention, such as a MAP, is not maintained, the treatment may fail. Modification of peptides is often needed to ensure that their activity can be maintained for hours rather than minutes [57]. This was the case with peptides W3R6 and eCAP, both of which saw marked improvement when made resistant to proteases through chemical modification [58,59]. This can also be achieved using drug delivery systems. Such delivery systems include polymers, lipid particles, and peptide-functionalized metallic nanoparticle complexes [60,61,62]. In these forms, the peptides are attached to a surface or inside of an existing membrane, thus preventing them from reaching the enzyme active site. In total, any intervention that could limit protease activity, without the need for synthetic peptide modification, would vastly improve peptide efficacity. The combinactivation of MAPSs, extracellular proteases inhibitors, and antimicrobials, as well as the synergy between them, is a topic to investigate that could lead to effective methods to treat bacterial and fungal infections.

### 3.2. In the Cell Wall and Outer Membrane

Bacteria are protected from attack not only by enzymes, but also by the structural components of their outer membrane and cell wall. These components are derived from polysaccharides, lipids, and proteins that make up a protective barrier to defend against entry into the cell and which occlude the cell membrane. Lipopolysaccharides (LPS) are a main component of bacteria cell walls of gram-negative bacteria and a significant drug target [63]. The LPS is hypothesized to interfere with MAPs via charge-charge interactions of the highly negatively charged LPS with cationic or amphiphilic MAPs [64]. Packing within the LPS layer is also crucially important as the same study showed that diastereomers of the cationic peptide KLLLKLKLKLLK showed a 20-fold difference in activity against *Escherichia coli*, *Shigella sonnei*, and *Salmonell enterica*. In addition to packing, specific LPS targeting sequences are important for any MAP to evade accumulation and degradation. A “boomerang” sequence, GWKRKRFG, was found to improve the MIC value of MAPs 20 to 50-fold against a panel of gram-positive and gram-negative bacteria [65]. Interrupting the binding between LPS and MAPs is, therefore, a crucial factor for MAP antimicrobial activity. Long-term retention by LPS prevents those peptides from ever reaching the cell membrane.

A second barrier to entry exists at the bacterial cell wall, especially in gram-positive bacteria, which consists of peptidoglycans and teichoic acids [66,67]. The peptidoglycan portion of the cell wall are glycans linked via d-amino acids and largely inhibits the transport of proteins larger than 50 kDa [68]. This size restriction, in general, does not inhibit MAPs, but can restrict peptides that aggregate before reaching the cell membrane. Moreover, anionic teichoic acids also present a challenge to MAPs. As previously discussed, lipopolysaccharide in gram-negative bacteria interacts with MAPs through charge-charge interactions. In a similar manner, teichoic acids of the cell wall in gram-positive bacteria can also drive charge-charge interactions with cationic or amphipathic MAPs [69]. In the case of both gram-positive and gram-negative bacteria, anionic structural molecules provide a level of resistance to MAPs that prevent them from reaching the cell membrane.

### 3.3. At the Cell Membrane

Once an intact MAP reaches the cell membrane it still needs to interact with either the surface or partition inside of the membrane/cell to achieve an antimicrobial effect. Biophysical studies of MAPs and claims of efficacy center on the idea that the lipid composition of the cell membrane stays relatively constant [70,71]. This research assumes that the composition of 10% charged lipid headgroups and 90% zwitterionic headgroups is appropriate to model a bacterial membrane. If a membrane contains this composition, it is assumed to be susceptible to MAPs and especially toward cationic MAPs [72]. To counter this assertion, there are emerging examples that suggest the cell membrane composition can be modified in response to membrane active antimicrobial peptides. 

There are examples of systems in microbes that exist solely to counter MAPs at the cell membrane. One such system is LiaFSR, which is a stress response system that is responsible for maintaining cell membrane integrity [73]. The system was characterized in *Enterococcus faecalis* but is conserved across Firmicutes. This system plays a key role in MAP resistance. Studying the effects of genetic mutation on its coding region showed that the protein LiaX, a previously unknown mediator to this response pathway, leads to hypersensitivity to daptomycin and other membrane targeting MAP-like peptides. The authors concluded that “LiaX functions as a modulator of the [cell membrane] stress response linking membrane adaptation, antibiotic resistance, and pathogenesis” [74]. The LiaX protein can mediate this important cellular function with its carboxy domain in intracellular space and responsible for activating the response. LiaX N-terminus is extracellular and is capable of sensing MAPs. It appears there is cooperativity in the binding of MAPs to extracellular N-terminus of the peptide. It is hypothesized that this is only one of a family of proteins that contributes to the stress response cascade of cell membrane remodeling [74]. Thus, there may be a widespread, broad-spectrum resistance mediating mechanism that MAPs will need to overcome in future mutant microbial strains.

More evidence has also arisen, showing that targeting cell membrane remodeling “flippase” proteins can help combat antimicrobial resistance. A class of antimicrobial MAP-like peptides called Humimycins was discovered in 2016 by a group at Rockefeller University using an in silico approach [75,76]. The reported MIC value of humimycin A(1) against the *S. aureus* strain *MRSA USA300* was 8µg/mL. At a lower concentration of 2µg/mL, humimycin was shown to potentiate β-lactam antibiotics, reducing the MIC value of carbenicillin to 1 µg/mL from 32 µg/mL. This effect is lost in mutants with a point mutation in the SAV1754 gene, which encodes for a homolog of a protein named MurJ. This protein, MurJ, is a flippase responsible for transporting lipid II from the inside of the cell to the outside. It is important to note that lipid II is the precursor molecule to peptidoglycan [77]. It follows that, in general, inhibition of the proteins responsible for lipid transport will impact the composition of cell walls and cell membranes. Active lipid flipping protects cells to some degree from mechanical damage and lysis caused by MAPs.

A significant final example comes from studies of the fungus *Cryptococcus neoformans*. This microorganism utilizes a P4-lipid flippase for, among other reasons, modulating the distribution of phosphatidyl serine (PS) lipids between the inner and outer leaflet [78,79]. Recent work by Xue and colleagues discovered a mutant strain of *C. neoformans* can be sensitized to the lipopeptide drug caspofungin [80]. They determined through genetic screening that the mutation was in the gene that encoded for Cdc50, a protein known to regulate the activity of P4-lipid flippase [81,82]. They further conducted experiments with fluorescently labeled PS to show that the concentration of the lipid on the outer membrane is significantly higher in mutants. They hypothesized that one of the main driving forces behind the sensitization of *C. neoformans* to caspofungin was this change in outer leaflet composition [83]. This provides an argument from reverse for the potential role of flippases in the development of resistance to MAP-like lipopeptide drugs or possibly to MAPs themselves. These previous examples demonstrate a correlation between sensitization to peptides and/or MAPs, and flippase activity that exists across many different species of bacteria and even into the fungi kingdom. Coupled with existing MAP-stress response systems; it is easy to see how cells can indeed evolve a method to dynamically change their cell membrane compositions to evade antimicrobial peptides. 

### 3.4. At the Point of Removal and/or Efflux

Peptides and small molecules that reach the cell membrane may still be removed, limiting their efficacy, and conferring cellular resistance. Efflux pumps are one of the most ubiquitous forms of antimicrobial-resistance, utilizing a transport system to expel antimicrobials from the cytoplasm or periplasmic space to lower the concentration below the MIC or effective level [84]. Bacterial efflux pumps are categorized into five major groups: ATP-binding cassette (ABC), major facilitator (MF), multidrug and toxic efflux (MATE), small multidrug resistance (SMR), and resistance nodulation division (RND) [85,86]. Apart from the ABC family, which operates through ATP hydrolysis, the other efflux families obtain their energy from the proton motive force. In particular, the RND family complex is in a wide range of Gram-negative and Gram-positive bacteria, consisting of a hydrophobic channel that can extrude multiple antimicrobial agents [87,88]. One way to counteract the antimicrobial-resistance via efflux pumps is through the incorporation of Efflux Pump Inhibitors (EPIs), however physiological cytotoxicity has been linked with EPIs when used in high dosages. To counteract this cytotoxicity, EPIs in combination with various antibiotics and MAPs have led to a synergistic effect that sensitizes bacteria to lower concentrations of antibiotics, MAPs, and EPIs [89,90]. Two of the most characterized EPIs used in research, and some clinical studies are 1-(1-naphthylmethyl)-piperazine (NMP) and l-Phe-l-Arg-β-napthylamide (PAβN) [91]. In recent studies, the addition of the MAP-like polymyxin B nonapeptide (PMBN) together with PAβN exhibited a synergistic effect and enhanced the antimicrobial activity of a spectrum of antibiotics by targeting the *MexAB-OprM* efflux pump constitutively expressed in strains of *Pseudomonas aeruginosa*. In combination with NMP, a similar effect was also seen; however, to a lesser extent than that in combination with PAβN [92]. Similarly, in another study with *P. aeruginosa* strains isolated from ICU patients, the combination of the MAP polymyxin E with PAβN indicated an increased production of *MexB* and *MexY* gene expression as a response in non-susceptible strains [93].

The *Sec* translocase is another receptor of the RND efflux pump family, often found in strains of *S. aureus* to transport cathelicidins, the host immune system MAPs. In a study examining *S. aureus* naturally displaying *SecDF* or engineered with a plasmid, the combination of NMP with the MAP CAP18 yielded lower MIC values. With the combination of PAβN and the MAP CAP18, similar results were achieved, but to a lesser degree. Similarly, some *S. aureus* strains tested in combination with the MAP LL-37 and NMP/PAβN presented with lower MIC values, following the same trend as that of CAP18 [94]. In strains of *E. coli* exhibiting another RND efflux pump, *AcrAB-TolC*, the combination of PAβN with various antimicrobial agents obtained from plants modeled a synergistic behavior with a decreasing trend in the MIC values as the concentration of PAβN increased [95]. This methodology of combining an efflux pump inhibitor with a MAP to combat antibiotic resistance is emerging in clinical fields, suggesting an important role for countering addressing efflux when working with MAP treatments [96,97].

## 4. Examples of Natural Synergies

### 4.1. Peptides and Mixtures

It is worth noting that many of the commonly used MAPs that are found in nature do not exist as pure, unaltered peptide extracts. The canonical antimicrobial MAP melittin, for instance, is isolated from a mixture of compounds of honey bee (*Apis mellifera*) venom [98,99,100]. Alone, melittin exhibits an MIC value ranging from 1–100 µg/mL against gram-positive and gram-negative bacteria [101]. Analysis of *Apis mellifera* venom extracts has shown that, while melittin may be the principal component, it is not the only factor that contributes to the MIC value [102]. A significant component is Mast Cell Degranulating Peptide (MCD), which is isolated from preparations of bee venoms [103,104]. This peptide is known to act by blocking fast-acting ion channels in mammalian cell membranes [105]. Another component protein, apamine, is also a channel blocking protein [106]. This mechanism of channel blocking adds potent neurotoxicity to apamine and MCD, but is largely confined to an effect on mammalian cells [107]. Evidence suggests that these peptides may form membrane-spanning pores, which further serve to destabilize microbial membranes [108]. Taken together, these peptides have a complementary effect, all forming structural changes in the membrane but minimal utility as membrane active antimicrobials.

One component of *Apis mellifera* venom makes a chemical change to the membrane itself: phospholipase A2 [109,110]. This protein is responsible for cleaving fatty acids from phospholipids, such as the type present in cell membranes [111]. Isolates of phospholipase A2 homologues from various sources were reported to exhibit MIC values in a range from 12.7–43.9 µg/mL against a panel of bacteria [112,113]. An interesting point to note is that the principal component analysis of honey bee venom identified phospholipase A2 as the main variable working synergistically with melittin to impact the overall MIC value [102]. The difference between apamine/MCD and phospholipase A2 is that while the former produces a structural rearrangement of phospholipids, the latter produces a chemical change. This chemical change leads to the creation of oxidative species and downstream toxic effects for microbes and mammalian cells alike, which limits their utility as a stand-alone antimicrobial agent [114,115]. In tandem, however, the ability of melittin and phospholipase A2 to sequester and then chemically modify microbial cell membranes at low concentrations could prove to be synergistic. 

It is reasonable to think that organisms evolved venoms as mixtures of different components specifically to combat the myriad of mechanisms for MAP resistance. Other homologues of phospoholipase A2 exist in the venoms of animals, such as snakes and wasps [116,117]. It is suggested that over the course of their evolution, various organisms incorporated the use of phospholipase A2 in their venoms for the cell membrane-damaging property [118]. Even certain fungi utilize phospholipase A2 as a means of attacking cellular membranes [119]. In any consideration of the key components of an antimicrobial attack, the importance of having both a membrane structure-altering component and a membrane chemistry-altering component cannot be ignored. The use of phospholipase molecules along with MAPs is an important source of potential natural synergy that warrants further synthetic investigation.

### 4.2. Peptide Prodrugs Susceptible to Proteolytic Enzymes

MAPs are expressed by cells first as prodrugs, which are later modified by peptidases to achieve full activity [120]. Different amino acid moieties are utilized both synthetically and naturally to limit the potential off-target activity of MAPs [121]. Promelittins, for example, need to be activated by proteolytic enzymes to create the functional, earlier discussed peptide: melittin [122,123]. Prodrug moieties act largely by altering the overall charge of the peptide, the hydrophobic moment, or by generally occluding the peptide. Altering the overall charge decreases membrane association with negatively charged bacterial membranes. This would inactivate cationic peptides. A change in the distribution of hydrophobic groups about the molecule would lead to an alteration in overall hydrophobic moment and disruption of membrane partitioning [124]. This would inhibit the activity of amphipathic α-helical MAPs like melittin. Finally, occlusion of the peptide, e.g., through self-association in the tertiary structure, would prevent the formation of pore or pore-like structures. All these modifications can severely inhibit peptide activity and raise the MIC or effectively make the peptides inactive towards microorganisms. 

Activation of MAP prodrugs by naturally expressed enzymes is crucial for their activity. The creation or utility of peptide prodrugs is an elegant strategy to evade the off-target MAP associated cytotoxic effects [125,126]. This is a form of naturally occurring synergy; peptide prodrugs and microbial enzymes work synergistically to activate the peptides. Clearly, the MIC value of the MAP and the concentration of the enzyme are directly correlated. While not a synergistic drug combination in the traditional sense, there is still an avenue for future research to improve this interaction. Panels of prodrug MAPs can be screened against known microbial proteolytic enzymes. These effects can also be compared with human plasma proteolytic enzymes to find moieties or existing prodrugs that are only cleaved in the presence of infectious agents. This type of strategy for synergy is also at the forefront of thinking with regards to MAPs. 

## 5. Conclusions

Our discussion of antimicrobial MAPs and the mechanisms of resistance may make the prospects for the field seem bleak. It is undeniable that there are ways in which cells can develop and have developed resistance to MAPs. Cells can degrade MAPs in the extracellular space. This can decrease the overall concentration below the minimum inhibitory concentration value for that peptide, causing less than full inhibition of growth. They can remodel their membrane to evade MAPs. This would directly increase the MIC value by making peptides less likely to bind to and interact with the membrane. Finally, they can remove MAPs from the membrane/intracellular space. Again, this can decrease the concentration in the membrane below an effective MIC and can also make the peptides more susceptible to the previous two methods of resistance.

Despite these facts, MAPs do offer significant advantages over small-molecule antimicrobial drugs. Small-molecule drugs have long struggled to overcome the double membrane of gram-negative bacteria [127,128]. Membrane active antimicrobial peptides are capable of directly disrupting the cell membrane, which circumvents this problem. This allows for a new way to treat existing antimicrobial-resistant infections [4]. Additionally, membrane disruptions allow for the intracellular delivery of antimicrobial agents, potentially re-sensitizing previously small-molecule resistant microorganisms. Of course, these benefits will not be achieved if microorganisms can resist MAPs themselves.

The way to counter resistance is to further develop these synergistic drug-MAP combinations. These were hinted at in studies of lipid flippases as described above, but also apply to combat extracellular proteases and efflux pumps. Inhibiting the degradation and removal of the peptide from the membrane space keeps the concentration at the appropriate minimum inhibitory concentration. Furthermore, we suggest that the synergistic interactions, inherent in venom mixtures, are a possible source to study synergies. Along with venoms, microbial enzymes themselves may provide synergy in the form of activating peptide pro-drugs. MAPs, as discussed in the literature, were never naturally intended to work alone. 

The closing point of this review is not to discourage the development or usage of MAPs as antimicrobial agents. Though there will always be a mechanism for cells to evade MAPs, there may also be ways to counter those mechanisms. A keen understanding of the balance between resistance and synergy is imperative for developing effective antimicrobial therapies. We hope to encourage mindfulness of MAP resistance exhibited by microbes and to excite further exploration of possible naturally occurring synergies that may already be at play in nature.

## Figures and Tables

**Figure 1 antibiotics-09-00620-f001:**
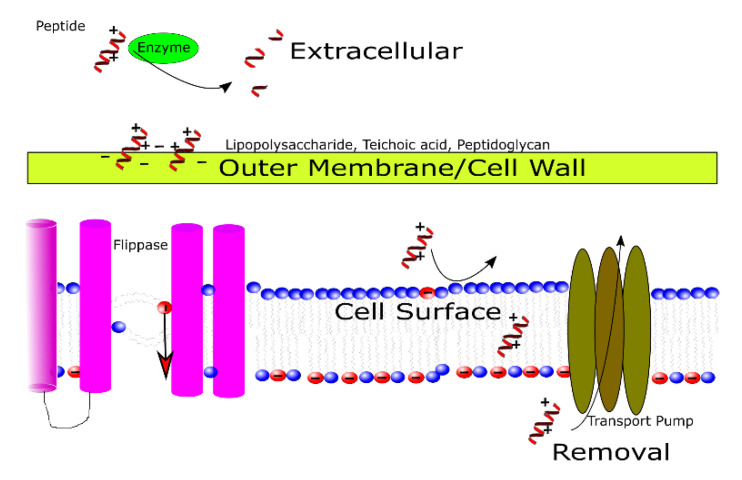
This figure provides a graphical overview of the information on possible methods by which cells may develop resistance to MAPs through protein structures, and hence, targets for synergistic interactions.

**Table 1 antibiotics-09-00620-t001:** Sequences of membrane-active peptides (MAPs)**.**

Peptide	Sequence
Melittin	GIGAVLKVLTTGLPALISWIKRKRQQ
Pixiganan	GIGKFLKKAKKFGKAFVKILKK
CAP18	GLRKRLRKFRNKIKEKLKKIGQKIQGLLPKLAPRTDY
LL-37	LLGDFFRKSKEKIGKEFKRIVQRIKDFLRNLVPRTES
K_5_L_7_	KLLLKLKLKLLK

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
