# Peer review of "Synergies with and Resistance to Membrane-Active Peptides"

_antibiotics, 2020, doi:10.3390/antibiotics9090620_

Round 1

Reviewer 1 Report

This manuscript "Synergies with and Resistance to Membrane Active Peptides" is submitted as a review in Antibiotics.

The review concerns specifically the antimicrobial peptides acting on the microbial membrane. The importance of the breakpoints introduced in the abstract is not really developed.

Three ways of resistance are presented: Outside of the Cell, At the Cell Membrane and At the Point of Removal and, or Efflux. Then examples of synergies are reported.

The review is well documented, however efflux is not very relevant for MAPs which are destabilizing the membrane rather to target intracellular components. On the other hand the relevant interactions of MAPs with bacterial wall, LPS or peptidoglycan are not at all presented. The interactions with the membrane are only view through the lipids, although all the cell wall components may be involved in the resistance acquisition. A small chapter should be introduced with some data.

Minor corrections

“A recently predicted minimal MAP unit LKDA was discovered which has a molecular weight of 445 Da[23]. The common antibiotic, Benzylpenicillin, has an MIC value of around 0.06 µg/mL and a molecular weight of 334 Da against certain bacteria strains[24]. Given this fact, and the fact that most MAPs will be larger than LDKA, it could be estimated that the efficacy of MAPs is at least two-fold and possibly more than ten-fold less than small molecule antibiotics on a mole-for-mole basis.”

There is not a real interest to self introduced this previous work in this part devoted to Defining Antimicrobial Synergy and Resistance.

L 274 prodrugs

Author Response

Thank you for your kind response.  We have addressed the critiques individual below and as noted within the text.

Reviewer 1

This manuscript "Synergies with and Resistance to Membrane Active Peptides" is submitted as a review in Antibiotics.

The review concerns specifically the antimicrobial peptides acting on the microbial membrane. The importance of the breakpoints introduced in the abstract is not really developed.

Three ways of resistance are presented: Outside of the Cell, At the Cell Membrane and At the Point of Removal and, or Efflux. Then examples of synergies are reported.

The review is well documented, however efflux is not very relevant for MAPs which are destabilizing the membrane rather to target intracellular components. On the other hand the relevant interactions of MAPs with bacterial wall, LPS or peptidoglycan are not at all presented. The interactions with the membrane are only view through the lipids, although all the cell wall components may be involved in the resistance acquisition. A small chapter should be introduced with some data.

We would like to thank the reviewer for this insight.   We’ve added an additional portion (new section 3.2) to discuss the impact of the LPS/peptidoglycan on peptides in both gram negative and gram positive bacteria.  We hope that the reviewer finds this section of value for the paper.  We also mention in the text the importance of MAPs in overcoming efflux of small molecules.  This section is more on the “synergy” side than the “resistance” side but we feel that it is still important to the manuscript overall.

See lines 189-215

Minor corrections

“A recently predicted minimal MAP unit LKDA was discovered which has a molecular weight of 445 Da[23]. The common antibiotic, Benzylpenicillin, has an MIC value of around 0.06 µg/mL and a molecular weight of 334 Da against certain bacteria strains[24]. Given this fact, and the fact that most MAPs will be larger than LDKA, it could be estimated that the efficacy of MAPs is at least two-fold and possibly more than ten-fold less than small molecule antibiotics on a mole-for-mole basis.”

There is not a real interest to self introduced this previous work in this part devoted to Defining Antimicrobial Synergy and Resistance.

We would like to thank the reviewers for this suggestion as well.  The LDKA peptide was specifically designed to be a minimal MAP unit and for this reason we initially included the reference.  We recognize, however, that it does appear a bit out of place given that no MIC value was established.  We therefore removed the reference and simply returned to the citation that talks about MAPs more generally.

See lines 79-103

L 274 prodrugs

fixed

Reviewer 2 Report

Kmeck et al. have written a nice review on membrane active peptides, with interesting discussions on how to circumvent resistance and synergise the activity of these promising drugs. I have some specific suggestions for improving the manuscript, which I trust will be useful:

  1. Abstract, line 10: “…challenge that notion”. It is not clear what “notion” is being referred to here.
  2. Fig 1 caption line 26: Should be “by which” rather than “why which”. Further, please label the figure more comprehensively (what are the pink, green, olive coloured structures, peptides are not labelled, etc.)
  3. Line 30: “…many, varied specific….” Please remove the word specific, it is implied already.
  4. Line 41: “Researchers in this area have repeated the claim that….” For such a strong statement, you must provide appropriate references.
  5. Line 55: Typo - “treat” should be “threat” I believe.
  6. In Section 2 (line 61 onwards), you use MICs as a standard for resistance levels, which is typical. However, with peptide antimicrobials, it is known that MICs are not ideal for quantifying efficacy (issues with inoculum effect etc). This is an important caveat to mention here. Please discuss briefly in this section the limitations of using MICs in studying peptides (for eg., see Jepson, A. K., Schwarz-Linek, J., Ryan, L., Ryadnov, M. G. & Poon, W. C. K. What Is the ‘Minimum Inhibitory Concentration’ (MIC) of Pexiganan Acting on Escherichia coli?—A Cautionary Case Study. in Leake M. (eds) Biophysics of Infection. Advances in Experimental Medicine and Biology, vol 915. Springer, Cham (2016)).
  7. Following the previous point, it would also be helpful to mention new biometrology assays to quantify peptide efficacy that avoid the use of MICs. For eg., to study the membranolytic efficacy of peptides, new lab-on-chip platforms have recently been developed (Al Nahas et al., Lab Chip 2019) that can provide model membranes of different compositions, on which peptides can be screened.
  8. Lines 70-71: the way the sentence is phrased is a bit odd, please put the molecular weight details before the MIC in the sentence.
  9. Lines 73-74 – I would like to reiterate my warning about using MICs to compare small molecule antibiotics versus peptides. As mentioned in the reference in point 6 above, the inoculum effect is a major technical problem in MIC measurements of peptides, more so than in the case of small molecule antibiotics. Suggest either adding this as a strong caveat here, or avoiding the comparison altogether.
  10. On the same point as above, considering the issues with standard MICs, it might be worth looking into the latest Antibiotic Susceptibility Testing assays (van Belkum et al. Nat Revs Microbiol. 2020 is a good recent review) to see whether any of those techniques could be used instead of MICs for peptides.
  11. Line 76: “The MIC values…” – please rephrase the sentence, the sentence structure is odd.
  12. Paragraph from Line 118 onwards: Could add a brief discussion about delivery systems to protect the peptides from degradation? For eg. Liposomal delivery?
  13. Line 150: Should be “able to sense…”
  14. Line 214: Should be “AcrAB-TolC” not ArcAB….
  15. Line 224: Typo (mellitin)
  16. Line 253: Typo (int-> in)
  17. Line 294: Typo (previous)
  18. Line 298: “remove” should be “removal”
  19. Line 306: “just as many” – this wording is not precise. I would suggest rephrasing. You cannot be sure that there are as many ways to circumvent resistance as there are mechanisms of resistance.
  20. Conclusions section – I found the first paragraph to be a bit too negative. There is one important advantage of membrane active peptides over small molecule antibiotics that has not been discussed at all in this paper. For gram-negative bacteria, which have complex double membranes, it is very difficult for small molecules to reach their intracellular target (see Silver, Bioorg Med. Chem. 2016, and Cama et al., J. Mol. Bio. 2019). Antibiotic discovery is challenging, and all types of antibiotics have to deal with resistance/tolerance problems. I would suggest discussing the conclusions in this more positive light, talking about how peptides circumvent the gram-negative membrane barrier problem of small molecules by directly attacking the membranes themselves. Then you can mention the issues with peptide based drugs, and how your review suggests ways to synergise their activity to make the most of this promising class of antibiotics.
  21. Following the previous point – there is a good recent review you could cite when discussing the importance of peptide based antibiotics in the conclusions (or earlier in the paper) – Magana et al., Lancet Infect. Dis., 2020.
  22. Finally, I picked up on some typos that I mentioned in the points above, but please go through the paper carefully before publication to check for spelling and sentence structure mistakes.  

Author Response

Thank you for your kind response.  We have addressed the critiques individual below and as noted within the text.

Kmeck et al. have written a nice review on membrane active peptides, with interesting discussions on how to circumvent resistance and synergise the activity of these promising drugs. I have some specific suggestions for improving the manuscript, which I trust will be useful:

  1. Abstract, line 10: “…challenge that notion”. It is not clear what “notion” is being referred to here.

We removed the line as it is confusion

  1. Fig 1 caption line 26: Should be “by which” rather than “why which”. Further, please label the figure more comprehensively (what are the pink, green, olive coloured structures, peptides are not labelled, etc.)

Fixed, figure is now better annotated.

See Lines 53-57

  1. Line 30: “…many, varied specific….” Please remove the word specific, it is implied already.

Fixed

See Line 31

  1. Line 41: “Researchers in this area have repeated the claim that….” For such a strong statement, you must provide appropriate references.

We provided references and also qualify this statement by saying “partially or totally” rather than stating that all researchers believe this to be the case.

See Line 48

  1. Line 55: Typo - “treat” should be “threat” I believe.

Fixed

  1. In Section 2 (line 61 onwards), you use MICs as a standard for resistance levels, which is typical. However, with peptide antimicrobials, it is known that MICs are not ideal for quantifying efficacy (issues with inoculum effect etc). This is an important caveat to mention here. Please discuss briefly in this section the limitations of using MICs in studying peptides (for eg., see Jepson, A. K., Schwarz-Linek, J., Ryan, L., Ryadnov, M. G. & Poon, W. C. K. What Is the ‘Minimum Inhibitory Concentration’ (MIC) of Pexiganan Acting on Escherichia coli?—A Cautionary Case Study. in Leake M. (eds) Biophysics of Infection. Advances in Experimental Medicine and Biology, vol 915. Springer, Cham (2016)).

Thank you for the suggestion, we will include this reference in the discussion and created a new section (2.2) critiquing the use of MIC assays and mentioning other assays that you have suggested.

See Lines 104-130

  1. Following the previous point, it would also be helpful to mention new biometrology assays to quantify peptide efficacy that avoid the use of MICs. For eg., to study the membranolytic efficacy of peptides, new lab-on-chip platforms have recently been developed (Al Nahas et al., Lab Chip 2019) that can provide model membranes of different compositions, on which peptides can be screened.

Agreed, see answer to point 6.

  1. Lines 70-71: the way the sentence is phrased is a bit odd, please put the molecular weight details before the MIC in the sentence.

Fixed

See Lines 88-90

  1. Lines 73-74 – I would like to reiterate my warning about using MICs to compare small molecule antibiotics versus peptides. As mentioned in the reference in point 6 above, the inoculum effect is a major technical problem in MIC measurements of peptides, more so than in the case of small molecule antibiotics. Suggest either adding this as a strong caveat here, or avoiding the comparison altogether.

Agreed, see answer to point 6.

  1. On the same point as above, considering the issues with standard MICs, it might be worth looking into the latest Antibiotic Susceptibility Testing assays (van Belkum et al. Nat Revs Microbiol. 2020 is a good recent review) to see whether any of those techniques could be used instead of MICs for peptides.

This is a good point.  We will include the review as a reference and mention briefly about the technique suggested above.  We still maintain that, given the widespread usage of MIC values, MICs warrant discussion as well.

See Lines 104-130

  1. Line 76: “The MIC values…” – please rephrase the sentence, the sentence structure is odd.

Changed to: Compared to β-lactams, there may be fewer mutant strains of microorganisms that can tolerate concentrations of membrane active peptides above their reported minimum inhibitory concentrations

See Lines 94-96

  1. Paragraph from Line 118 onwards: Could add a brief discussion about delivery systems to protect the peptides from degradation? For eg. Liposomal delivery?

Added: This can also be achieved through the use of drug delivery systems.  These delivery systems include polymers, lipid particles, and peptide-functionalized metallic nanoparticle complexes [55–57].  In these forms, the peptides are attached to a surface or inside of an existing membrane, thus preventing them from reaching the enzyme active site.

Lam, S.J.; O’Brien-Simpson, N.M.; Pantarat, N.; Sulistio, A.; Wong, E.H.H.; Chen, Y.Y.; Lenzo, J.C.; Holden, J.A.; Blencowe, A.; Reynolds, E.C.; et al. Combating multidrug-resistant Gram-negative bacteria with structurally nanoengineered antimicrobial peptide polymers. Nat. Microbiol. 2016, 1, doi:10.1038/nmicrobiol.2016.162.

Huang, C.; Jin, H.; Qian, Y.; Qi, S.; Luo, H.; Luo, Q.; Zhang, Z. Hybrid melittin cytolytic peptide-driven ultrasmall lipid nanoparticles block melanoma growth in vivo. ACS Nano 2013, 7, 5791–5800, doi:10.1021/nn400683s.

Zong, J.; Cobb, S.L.; Cameron, N.R. Peptide-functionalized gold nanoparticles: Versatile biomaterials for diagnostic and therapeutic applications. Biomater. Sci. 2017, 5, 872–886, doi:10.1039/c7bm00006e.

See Lines 180-183

  1. Line 150: Should be “able to sense…”

We are unclear about the reviewer’s intentions, we changed to “is capable of sensing membrane active peptides”

See line 235

  1. Line 214: Should be “AcrAB-TolC” not ArcAB….

Fixed

  1. Line 224: Typo (mellitin)

Fixed

  1. Line 253: Typo (int-> in)

Fixed

  1. Line 294: Typo (previous)

Fixed

  1. Line 298: “remove” should be “removal”

Fixed

  1. Line 306: “just as many” – this wording is not precise. I would suggest rephrasing. You cannot be sure that there are as many ways to circumvent resistance as there are mechanisms of resistance.
  2.  

We agree that this is a fair point, we changed it to “also ways”.

  1. Conclusions section – I found the first paragraph to be a bit too negative. There is one important advantage of membrane active peptides over small molecule antibiotics that has not been discussed at all in this paper. For gram-negative bacteria, which have complex double membranes, it is very difficult for small molecules to reach their intracellular target (see Silver, Bioorg Med. Chem. 2016, and Cama et al., J. Mol. Bio. 2019). Antibiotic discovery is challenging, and all types of antibiotics have to deal with resistance/tolerance problems. I would suggest discussing the conclusions in this more positive light, talking about how peptides circumvent the gram-negative membrane barrier problem of small molecules by directly attacking the membranes themselves. Then you can mention the issues with peptide based drugs, and how your review suggests ways to synergise their activity to make the most of this promising class of antibiotics.

Thank you for these insights.  We will retain the original structure of the conclusion section as a stylistic choice but will highlight some of the important aspects of membrane targeting that have been discussed by others and how it relates to the issues involved in small molecule targeting of gram negative bacteria.

See lines 370-403

  1. Following the previous point – there is a good recent review you could cite when discussing the importance of peptide based antibiotics in the conclusions (or earlier in the paper) – Magana et al., Lancet Infect. Dis., 2020.

We will add this review as well.

See lines 28-29

  1. Finally, I picked up on some typos that I mentioned in the points above, but please go through the paper carefully before publication to check for spelling and sentence structure mistakes.  

Thank you for your kind and thorough review of the manuscript, we will make sure to correct those mistakes and others.

Reviewer 3 Report

In this review article by Kmeck et al. to ‘Antibiotics’, the authors provide a focused and concise overview of the recent scientific advancements and current understanding of antimicrobial resistance (AMR) to membrane-active peptides (MAPs) as well as possible ways to overcome this issue. This information could be beneficial for broad scientific communities working in the fields of antimicrobial resistance and drug development. Some aspects absent in this manuscript could improve this review to make it more comprehensive.

I think more information should be added to the MAPs' definition and function. Another issue that might be highlighted is whether glycopeptides and lipopeptides are also considered as MAPs.

Figure 1 should be placed closer to its first mention in the text. In addition, it could be better structured and described "what is what".

29. Maybe more examples on the peptides with the references to original works?

77. I don't think that there is a special problem with the breakpoints definition for the MAPs. The matter only in how widely a drug is used, not in its mechanism (daptomycin has clinical breakpoints for S. aureus and Streptococcus spp). The breakpoints are determined uniformly in vitro, and hence it is not subject to an in vivo degradation by the host. To analyze AMR development, one can perform corresponding long-time experiments by observing an increase in MIC.

277. "peptide prodrugs and microbial enzymes work synergistically to create an overall response" - please, explain. Response to what? to microbes?

Some discussion on the role of MAPs on host immunity modulation would help understand how this may affect overall response to microbes.

Minor points:
64. Fractional MIC Index (FIC) - abbreviation is not correct.
69. LKDA?
75. "MAPs are less susceptible to resistance" I understand what you mean, but it sounds incorrect, it might be better to re-arrange this sentence.
94. some fungal species or most of them?
260. please clarify, they CAN BE expressed or expressed?
Some typos are present through the text, which needs to be carefully checked.

Author Response

Thank you for your kind response.  We have addressed the critiques individual below and as noted within the text.

In this review article by Kmeck et al. to ‘Antibiotics’, the authors provide a focused and concise overview of the recent scientific advancements and current understanding of antimicrobial resistance (AMR) to membrane-active peptides (MAPs) as well as possible ways to overcome this issue. This information could be beneficial for broad scientific communities working in the fields of antimicrobial resistance and drug development. Some aspects absent in this manuscript could improve this review to make it more comprehensive.

I think more information should be added to the MAPs' definition and function. Another issue that might be highlighted is whether glycopeptides and lipopeptides are also considered as MAPs.

Greater detail was added to discuss structural characteristics of MAPs as well as a table showing the sequences of MAPs discussed.  The authors would argue that certain glycopeptides and lipopeptides may be considered MAP-like when one of their main role sis to interact with the cell membrane.  An example of this would be daptomycin, which forms disruptions of the cell membrane. In general though they are not considered to be “membrane active peptides” in the traditional sense.  Relevant examples in the text will be highlighted to note the difference.  Even though an antimicrobial may be a “lipopeptide” or a “MAP” the resistance mechanisms highlighted may work in a similar manner against both and are provided as examples.

See lines 20-77

Figure 1 should be placed closer to its first mention in the text. In addition, it could be better structured and described "what is what".

We have moved the figure closer to the paragraph in which it is mentioned and also provided improved captions to help the readers better understand the figure.

See Lines 53-57

  1. Maybe more examples on the peptides with the references to original works?

Table 1 provides some examples, the references for those peptides can be found within the text.

See Lines 40-41

  1. I don't think that there is a special problem with the breakpoints definition for the MAPs. The matter only in how widely a drug is used, not in its mechanism (daptomycin has clinical breakpoints for S. aureusand Streptococcus spp). The breakpoints are determined uniformly in vitro, and hence it is not subject to an in vivodegradation by the host. To analyze AMR development, one can perform corresponding long-time experiments by observing an increase in MIC.

See added discussion on MIC values (section 2.2)  Another reviewer raised some questions about the validity of MIC values and we have added this point.  We agree that analysis of changes in MIC values over time is important but have added a discussion of the complications related to peptides to further explore the issue.

See Lines 104-130

  1. "peptide prodrugs and microbial enzymes work synergistically to create an overall response" - please, explain. Response to what? to microbes?

We have changed it to “work synergistically to activate the peptides.”

See lines 361-362

Some discussion on the role of MAPs on host immunity modulation would help understand how this may affect overall response to microbes.

We incorporated information in the text to discuss the apparent anti-inflammatory role of MAPs

See Lines 161-171

Minor points:
64. Fractional MIC Index (FIC) - abbreviation is not correct.

Fixed

See Lines 82-83

  1. LKDA?

Removed

  1. "MAPs are less susceptible to resistance" I understand what you mean, but it sounds incorrect, it might be better to re-arrange this sentence.

This was changed in the text

See Lines: 93-96

  1. some fungal species or most of them?

We removed the term “some” and replaced it with “certain”

See Lines 137-138

  1. please clarify, they CAN BE expressed or expressed?

changed to “are expressed as prodrugs”

See Line 345

Some typos are present through the text, which needs to be carefully checked.

We have reviewed the text for typos.

Round 2

Reviewer 1 Report

This revised manuscript "Synergies with and Resistance to Membrane Active Peptides" corresponds to the expected correction but the new section 3.2.

Overall, the review has been improved and should be soon accepted.

Nevertheless, the role of LPS in the peptide interaction of MAPs is described as a trap for aggregated peptides according to the introduced references 65 and 66. However, this aspect associated with peptide aggregation corresponds to only one aspect of the LPS role. A more subtle role of LPS is often described. The core oligosaccharide is involved in the tuning of the peptide interaction. After binding the lateral diffusion of the LPS is enhanced by displacement of dicationic ions allowing direct interaction of the MAP with the lipidic part.

One recent article (see below) describes this aspect.

Ebbensgaard, A., Mordhorst, H., Aarestrup, F. M., & Hansen, E. B. (2018). The role of outer membrane proteins and lipopolysaccharides for the sensitivity of escherichia coli to antimicrobial peptides. Frontiers in Microbiology, 9. https://doi.org/10.3389/fmicb.2018.02153

Packing aggregated peptides is not the only role of LPS, it should be introduced in the section 3.2.